# Translation and cultural adaptation of MedStopper®—A web-based decision aid for deprescribing in older adults: A protocol

**Luís Monteiro**[1,2]*, **Sofia Baptista**[1,3], **Inês Ribeiro-Vaz**[1,3,4], **James McCormack**[5], **Cristiano Matos**[6], **Andreia Teixeira**[1,3,7], **Matilde Monteiro-Soares**[1,3], **Carlos Martins**[1]

1 CINTESIS@RISE—Centre for Health Technology and Services Research, Faculty of Medicine, University of Porto, Porto, Portugal, 2 Department of Medical Sciences, University of Aveiro, Aveiro, Portugal, 3 MEDCIDS—Department of Community Medicine, Information and Decision in Health, Faculty of Medicine, University of Porto, Porto, Portugal, 4 Porto Pharmacovigilance Centre, Faculty of Medicine, University of Porto, Porto, Portugal, 5 Faculty of Pharmaceutical Sciences, University British Columbia, Vancouver, Canada, 6 Escola Superior de Tecnologia da Saúde de Coimbra, Instituto Politécnico de Coimbra, Coimbra, Portugal, 7 IPVC—Instituto Politécnico de Viana do Castelo, Viana do Castelo, Portugal

* luismonteiro.net@gmail.com

**Data Availability Statement:** All relevant data are within the paper.

## Abstract

### Background

Older patients are more likely to have medication-related problems, which are associated with changes in pharmacokinetics and pharmacodynamics, multimorbidity, and polypharmacy. Polypharmacy and inappropriate prescribing are well-known risk factors which commonly cause adverse clinical outcomes in older people. Prescribers struggle to identify potentially inappropriate medications and to choose an adequate tapering approach.

### Methods/design

The goal of the study is to translate and culturally adapt MedStopper®, an original English language web-based decision aid system in deprescribing medication, to the Portuguese population. A translation-back translation method, with validation of the obtained Portuguese version of MedStopper® will be used, followed by a comprehension test.

### Discussion

This is the first research in the Portuguese primary care setting that aims to provide a useful online tool for the appropriate prescription of older patients. The translated version in Portuguese version of the MedStopper® tool will represent an advance that seeks to continue improving the management of medications in the elderly. The adaptation into Portuguese of the educational tool provides clinicians with a screening tool to detect potentially inappropriate prescribing in patients older than 65 that reliable and easier to use.

### Trial registration

Retrospectively registered.

**Funding:** Fundação para a Ciência e Tecnologia, I. P. within CINTESIS, R&D Unit (reference UIDB/4255/2020). The funders had no role in study design, data collection and analysis, decision to publish, or preparation of the manuscript."

**Competing interests:** The authors have declared that no competing interests exist.

**Abbreviations:** PIMs, potentially inappropriate medications; STOPP, Screening Tool of Older People's Prescriptions; START, Screening Tool to Alert to Right Treatment; ECDC, European Centre for Disease Prevention and Control's.

# Background

Older adults are more likely to have medication-related problems, which are associated with changes in pharmacokinetics and pharmacodynamics, multimorbidity [1], and polypharmacy [2]. Polypharmacy can be defined as taking more than five medications simultaneously [3]. The prevalence of this phenomenon can range from 30 to 70% [4], but in Portugal, it is estimated to reach up to 77% of the elderly population [5].

Deprescribing is the process of stopping or tapering an inappropriate medication [6]. It is not just a way of managing polypharmacy; it has the clear goal of improving patient outcomes.

Despite the importance of adequate deprescribing, there are several barriers to its success, such as patient reluctance, limited time, medical, cultural inertia, and difficulty in recognizing potentially inappropriate medications [PIMs] [7].

PIMs can be described as the use of medications that potentially have more risks than benefits even though safer pharmacologic and nonpharmacologic alternatives are available [8].

These PIM are listed in several clinical tools, for example, by the EU PIM-List [9], Beers Criteria [10], and the Screening Tool of Older People's Prescriptions (STOPP) and the Screening Tool to Alert to Right TreatmentSTART) [11]. Nevertheless, it is hard to recognize inappropriate medications in individual patients [7], and computerized decision support tools may help to address polypharmacy [12], reduce the number of potentially inappropriate prescriptions started and the mean number of potentially inappropriate prescriptions per patient [13]. For all of this, having a computerized decision support tool that includes both Beers criteria and STOPP/START could enhance deprescribing.

The MedStopper® is an excellent example of such tool being a web tool that integrates both Beers criteria and STOPP/START. It was developed to help clinicians and patients to make decisions about reducing or stopping medications. By entering the list of medications, a patient is receiving, MedStopper® sequences the drugs from "more likely to stop" to "less likely to stop", based on three key criteria: the potential of the drug to improve symptoms, its potential to reduce the risk of future illness, and its likelihood of causing harm. Three rating levels under each of the ranking categories: Unhappy Face, Neutral Face, and Happy Face. Suggestions for how to taper the medication are also provided and possible symptoms when stopping or tapering. The MedStopper® is available in two languages: English and French [14]. At first, the user is asked to identify if the patient is a frail elderly; after that, the user can search for medications by generic or brand name; in the next step, the user is asked to select the condition for which the medication is used. The MedStopper® Plan will display the medications in order of stopping priority and in colors: red is the highest stopping priority and green the lowest stopping priority.

We aim to translate and culturally adapt MedStopper®, an original English language web-based decision aid system in deprescribing medication, to the Portuguese population [14].

# Methods

## The conceptual framework for adaptation

We will follow the guidelines provided by the European Centre for Disease Prevention and Control's (ECDC) [15] for the cultural adaptation of health materials. Therefore, we will follow their five-step approach to guarantee that we will not only translate MedStopper® but to ensure quality, comprehension, contextual and cultural appropriateness, and applicability (Timeline available in S1 Timeline Appendix).

**Step 1: Selection of materials and process coordinators.** We searched online throughout professional support groups, such as NSW Therapeutic Advisory Group [16], the Canadian

Deprescribing Network [17], Primary Health Tasmania [18], and the US Deprescribing Research Network [19], to find free access online web-based tools for the identification of potentially inappropriate medications identified by the STOPP/START criteria and the Beers criteria. We identified the MedStopper® tool [14].

We contacted one of the original authors [JM which is also an author of this protocol], and the adaptation of the tool for the Portuguese population was agreed. We will review the original version of the tool and, in cooperation with its developers; we will identify the tool's main elements. We will identify the potentially inappropriate medications, a user-friendly format [icons that instantly give feedback], a rationale and summary of the STOPP/START, and Beers criteria and a deprescribing plan. LM will be the process coordinator.

**Step 2: Early review.** The translation will only take place after the revision of the tool by the process coordinator [LM] and by a linguistic expert native speaker of English and fluent speaker of Portuguese.

The linguistic expert will do the translation after contact with one of the developers of the original tool. This collaboration will make sure that difficult concepts will be identified and alternative ways to express the information are found.

Examples of predictable changes after this early review include incorporating recommendations of Portuguese Directorate General of Health; and adding a calendar that visually shows the daily dose of the medicines during the chosen deprescribing process. Also, in 2019, the Beers criteria were reviewed, therefore, we will update the MedStopper® tool accordingly.

**Step 3: Translation and back-translation.** This step will not be just a literal translation of the content; in this third step, we will make the translation, quality check, and an independent review. We will then have two forward translations done by two professional translators, expert native speakers of English and fluent speakers of Portuguese. From these two translations, we will obtain a consensual translated version considering the debate between the translators and the research team.

The next step will then be to obtain a back-translated version by a professional translator, native speaker of English, fluent in Portuguese. An independent expert, who also understands the original language, will review the final version of the tool.

**Step 4: Comprehension testing.** To guarantee that the adapted and translated decision tool is clear for the Portuguese doctors, we will conduct this step. According to the ECDC [15], one of the possible approaches is to have individual interviews. We will perform interviews in which participants are asked, while using the web-based decision-aid, to express their opinion about the content and usability. All participants will be asked to apply the MedStopper® tool to the fictitious clinical case that was previously tested by family doctors with a special interest in elderly care: First clinical case, with a frail elderly patient taking alprazolam for insomnia, omeprazole for heartburn, gliclazide for type 2 diabetes, naproxen for osteoarthritis and aspirin for the previous stroke; second clinical case, a frail elderly patient with hypertension and diabetes taking furosemide, ramipril/amlodipine, metformin, atorvastatin, trospium chloride and recently medicated with donepezil.

Eligible participants family doctors will be recruited through an invitation to participate by mailing lists and social media. Candidates who respond to the email will be invited for individual semi-structured interviews to take place in the Faculty of Medicine of the University of Porto or another place of convenience for the participant, provided that confidentiality is assured. All participants will receive written information about the study and will sign individual consent forms. We plan a convenience sample of participants until data saturation is reached, which means the collection of qualitative data will be made to the point of closure is attained because new data is considered redundant information [20]. To ensure confidentiality, each interview will be given an alphanumeric code.

Three researchers will conduct the interviews following an interview guide. We will conduct a pilot test to make sure that the interview guide is clear. After gathering the written consent by the participants, they will be asked to give demographic and professional data such as age, number of clinical years, sex, and whether they are interns or specialists in family medicine.

The individual interviews will be audio-recorded, and the researchers will apply cognitive methods such as think aloud, and paraphrasing techniques [21].

Each participating doctor will have a computer with access to the web-based format of the decision aid tool—MedStopper®. They will apply the tool to the same clinical cases presented to the participants. For each clinical scenario, a list of medications and diseases for fiction elderly patients will be presented, and the participants will be encouraged to apply the Med-Stopper® tool.

Participants will be encouraged to share their opinion aloud while they are going through the tool to identify potential issues in the format and content [22]. At the end of the interview, we intend to use a probing technique retrospectively with questions to elicit how doctors felt throughout the process.

Data obtained after verbatim transcription of the audio files will be analyzed in a personal computer [without a network connection] belonging to one of the researchers.

One month after data analysis, the audio files will be destroyed. Using Ligre® Software, we will conduct a thematic categorical analysis.

Two authors will independently analyze the transcripts and fill a content analysis form. The participants' feedback will be categorized into six main themes: navigation and usability of the interface, content comprehension and completeness, length of the decision aid tool and amount of information, use of figures/illustrations, time of completion [23], and other issues [e.g., suggestions to improve the decision aid tool]. We will include participants' proposed solutions or revisions.

**Step 5: Proofreading.**   Two Portuguese native speaker doctors, selected by the process coordinator, will conduct the final step to check for minor errors which have been missed during the translation process.

## Ethics approval and consent to participate

Every participant will sign a written consent form. The identity of all participants will be protected throughout the study.

This protocol was approved on 23rd April 2021 by the Health Ethics Committee from Centro Hospitalar de São João/Faculdade de Medicina da Universidade do Porto with the reference number 134/21.

Results from this study will be disseminated in peer-reviewed publications, conference presentations and reports.

## Discussion

This study will allow the adequate use of the MedStopper® tool in the Portuguese population. The translated version in Portuguese version of the MedStopper® tool will represent an advance that seeks to continue improving the management of medications in the elderly. The adaptation into Portuguese of the educational tool provides clinicians with a screening tool to detect potentially inappropriate prescribing in patients older than 65 that is reliable and easier to use.

This tool will also be useful to improve medical education. In fact, portuguese speaking students and junior doctors may practice on clinical case vignettes by applying the MedStopper® tool.

This is the first research in the Portuguese primary care setting that aims to provide a useful online tool for the appropriate prescription of older patients.

The main limitation of this study is that it will be conducted in Portugal, and it may not apply to other Portuguese-speaking countries.

## Supporting information

**S1 Timeline.**
(PDF)

## Author Contributions

**Conceptualization:** Luís Monteiro, James McCormack, Andreia Teixeira, Carlos Martins.

**Formal analysis:** Luís Monteiro.

**Investigation:** Sofia Baptista, Inês Ribeiro-Vaz.

**Methodology:** Sofia Baptista, Inês Ribeiro-Vaz.

**Writing – original draft:** Luís Monteiro, Andreia Teixeira, Matilde Monteiro-Soares, Carlos Martins.

**Writing – review & editing:** Luís Monteiro, Cristiano Matos, Andreia Teixeira, Matilde Monteiro-Soares, Carlos Martins.

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
