## [Decision Letter · Decision Letter 0]

11 Jan 2023

PONE-D-22-30219PLOS ONE Translation and cultural adaptation of a web-based decision aid for deprescribing in older adults: A protocolPLOS ONE

Dear Dr. Monteiro,

Thank you for submitting your manuscript to PLOS ONE. After careful consideration, we feel that it has merit but does not fully meet PLOS ONE’s publication criteria as it currently stands. Therefore, we invite you to submit a revised version of the manuscript that addresses the points raised during the review process.

We look forward to receiving your revised manuscript.

Kind regards,

Simon White

Academic Editor

PLOS ONE

Journal Requirements:

“Fundação para a Ciência e Tecnologia, I.P. within CINTESIS, R&D Unit (reference UIDB/4255/2020).”

Reviewers' comments:

Reviewer's Responses to Questions

**Comments to the Author**

1. Does the manuscript provide a valid rationale for the proposed study, with clearly identified and justified research questions?

Reviewer #1: Yes

Reviewer #2: Yes

2. Is the protocol technically sound and planned in a manner that will lead to a meaningful outcome and allow testing the stated hypotheses?

Reviewer #1: Yes

Reviewer #2: Yes

3. Is the methodology feasible and described in sufficient detail to allow the work to be replicable?

Reviewer #1: Yes

Reviewer #2: Yes

4. Have the authors described where all data underlying the findings will be made available when the study is complete?

Reviewer #1: Yes

Reviewer #2: No

5. Is the manuscript presented in an intelligible fashion and written in standard English?

Reviewer #1: Yes

Reviewer #2: Yes

6. Review Comments to the Author

You may also provide optional suggestions and comments to authors that they might find helpful in planning their study.

Reviewer #1: 5. Is the manuscript presented in an intelligible fashion and written in standard English?

There are some fixes needed:

In the fifth paragraph of the Background, the EU(7)-PIM List is written as "EU (7) PIM-list" which is confusing, as "(7)" appears to be a reference.

In the first paragraph of the Discussion, "... older than 65 that reliable and ..." seems to be missing the verb.

In the second paragraph of the Discussion, instead of "useful for improve" it should be "useful for improving" or "useful to improve".

In the References, all the references with a website (14, 16, 17, 18, 19) are missing the last "]".

Reviewer #2: Review

Thank you for the opportunity to review this manuscript. I enjoyed reading the article, mainly because this study protocol aims to meet Portuguese physicians' needs in daily clinical practice by supporting/facilitating the deprescribing process.

The study protocol aims to translate and culturally adapt a web-based decision aid system for deprescribing to European Portuguese: the MedStopper®. This tool is free and supports the physician in identifying the potentially inappropriate medications (PIM) based simultaneously on Beer's criteria and STOPP/START tool. Additionally, this tool sequences PIM according to their deprescribing priority, suggest how to reduce or stop each medication, and informs about the potential adverse drug withdrawal events and how to prevent or manage them.

The abstract provides a concise and clear overview of the study protocol.

Overall, the paper has a logical flow and good intelligibility.

Major issues:

- No major issues were found.

Minor issue:

a) Background

The background section focuses on relevant recent research to support this study.

- Minor recommendation

It explains why the study was necessary, but in my opinion, it could be more explicit that Medstopper® is a tool that will address Portuguese clinicians' needs. First, the authors identified physicians' needs very clearly: "For all of this, having a computerized decision support tool that includes both Beers criteria and STOPP/START could enhance deprescribing". Then, in a different paragraph, the authors describe the Medstopper®. Still, it is not clear to readers unfamiliar with this tool that it will address the need identified in the previous paragraph: to include both Beers and STOPP/START criteria. Maybe a simple connection phrase or expression will make it clear.

b) Methods

The study protocol design and methodology used are appropriate to reach the aims presented in the background.

- Minor recommendation (discretionary)

Consider publishing the semi-structured interview guide. This recommendation is optional, as the authors may choose to publish the interview guide in the research results paper.

7. PLOS authors have the option to publish the peer review history of their article (what does this mean?). If published, this will include your full peer review and any attached files.

Reviewer #1: No

Reviewer #2: **Yes: **Anabela Inácio Pereira

---

## [Author Response · Author response to Decision Letter 0]

9 Mar 2023

Editorial Comments

Journal Requirements:

Response: We thank you for your comment. We edited the manuscript makinh sure it folloew PLOS ONE style template.

Response: We thank you for your remark. We edited this information in order to macth the “Funding Information’ and ‘Financial Disclosure’ sections

“Fundação para a Ciência e Tecnologia, I.P. within CINTESIS, R&D Unit (reference UIDB/4255/2020).”

Response: We thank you for your remark. In fact the funders had no role. Therefore we added the following statement (also in the cover letter)

Response: We thank you for your remark. In response to your comments, we would like to clarify that the data underlying the results described in our manuscript is permanently available online at https://medstopper.com/. We understand the importance of making data available to the scientific community and fully support the goal of open and transparent research. The data used in our study is publicly accessible through the website and does not contain any potentially identifying patient information, which has been fully anonymized. The patient data collected from the comprehension test will be made available at the end of the study. The data will be kept confidential and anonymous to protect patient privacy. Access to the data will be granted to qualified researchers upon request, subject to any ethical and legal restrictions.

The following statement was added in the manuscript: “Data Availability:All relevant data are within the paper”

Response: This is not applicable to our study. 

Response: This is not applicable to our study. 

Response: Thank you for your feedback on our manuscript. In response to your comments, we would like to clarify that the data underlying the results described in our manuscript is permanently available online at https://medstopper.com/. We understand the importance of making data available to the scientific community and fully support the goal of open and transparent research. The data used in our study is publicly accessible through the website and does not contain any potentially identifying patient information, which has been fully anonymized. The patient data collected from the comprehension test will be made available at the end of the study. The data will be kept confidential and anonymous to protect patient privacy. Access to the data will be granted to qualified researchers upon request, subject to any ethical and legal restrictions.

Response: We thank you for your comment. The ethics statement is now only in the Methods section.

Response: We thank you for your remark. We reviewed the reference list.

 

External Peer-Review Report(s)

Reviewer: 1

Response: We thank you for your comments and inputs that made improved this manuscript.

General comments

5. Is the manuscript presented in an intelligible fashion and written in standard English?

There are some fixes needed:

In the fifth paragraph of the Background, the EU(7)-PIM List is written as "EU (7) PIM-list" which is confusing, as "(7)" appears to be a reference.

Response: We thank you for your remark. We edited this phrase to the following:

“by the EU PIM-List (9), Beers Criteria”

In the first paragraph of the Discussion, "... older than 65 that reliable and ..." seems to be missing the verb.

Response: We thank you for your comment. We edited this phrase to the following:

“prescribing in patients older than 65 that is reliable and easier to use.”

In the second paragraph of the Discussion, instead of "useful for improve" it should be "useful for improving" or "useful to improve".

Response: We thank you for your suggestion. We changed the sentence to:

“This tool will also be useful to improve medical education”

In the References, all the references with a website (14, 16, 17, 18, 19) are missing the last "]".

Response: We thank you for your remark. We reviewed the reference list and added the last "]".

 

Reviewer: 2

General comments

Reviewer #2: Review

Thank you for the opportunity to review this manuscript. I enjoyed reading the article, mainly because this study protocol aims to meet Portuguese physicians' needs in daily clinical practice by supporting/facilitating the deprescribing process.

The study protocol aims to translate and culturally adapt a web-based decision aid system for deprescribing to European Portuguese: the MedStopper®. This tool is free and supports the physician in identifying the potentially inappropriate medications (PIM) based simultaneously on Beer's criteria and STOPP/START tool. Additionally, this tool sequences PIM according to their deprescribing priority, suggest how to reduce or stop each medication, and informs about the potential adverse drug withdrawal events and how to prevent or manage them.

The abstract provides a concise and clear overview of the study protocol.

Overall, the paper has a logical flow and good intelligibility.

Response: We thank you for your comments and inputs that allowed us to improve this manuscript.

Have the authors described where all data underlying the findings will be made available when the study is complete?

Response: We thank you for your remark. We added informations describing where all data will be available.

Response: Thank you for your feedback on our manuscript. In response to your comments, we would like to clarify that the data underlying the results described in our manuscript is permanently available online at https://medstopper.com/. We understand the importance of making data available to the scientific community and fully support the goal of open and transparent research. The data used in our study is publicly accessible through the website and does not contain any potentially identifying patient information, which has been fully anonymized. The patient data collected from the comprehension test will be made available at the end of the study. The data will be kept confidential and anonymous to protect patient privacy. Access to the data will be granted to qualified researchers upon request, subject to any ethical and legal restrictions.

Major issues:

- No major issues were found.

Minor issue:

a) Background

The background section focuses on relevant recent research to support this study.

- Minor recommendation

It explains why the study was necessary, but in my opinion, it could be more explicit that Medstopper® is a tool that will address Portuguese clinicians' needs. First, the authors identified physicians' needs very clearly: "For all of this, having a computerized decision support tool that includes both Beers criteria and STOPP/START could enhance deprescribing". Then, in a different paragraph, the authors describe the Medstopper®. Still, it is not clear to readers unfamiliar with this tool that it will address the need identified in the previous paragraph: to include both Beers and STOPP/START criteria. Maybe a simple connection phrase or expression will make it clear.

Response: We thank you for your comment. We edited the paragraph to the following

“ The MedStopper® is an excellent example of such tool being a web tool that integrates both Beers criteria and STOPP/START. It was developed to help clinicians and patients to make decisions about reducing or stopping medications”

b) Methods

The study protocol design and methodology used are appropriate to reach the aims presented in the background.

- Minor recommendation (discretionary)

Consider publishing the semi-structured interview guide. This recommendation is optional, as the authors may choose to publish the interview guide in the research results paper.

Response: We thank you for your comment. We choose to publish the interview guide in the research results paper and not on the protocol.

---

## [Editor Report · Decision Letter 1]

3 Apr 2023

Translation and cultural adaptation of MedStopper® - a web-based decision aid for deprescribing in older adults: a protocol

PONE-D-22-30219R1

Dear Dr. Monteiro,

We’re pleased to inform you that your manuscript has been judged scientifically suitable for publication and will be formally accepted for publication once it meets all outstanding technical requirements.

Kind regards,

Simon White

Academic Editor

PLOS ONE
---

## [Editor Report · Acceptance letter]

6 Apr 2023

PONE-D-22-30219R1 

Translation and cultural adaptation of MedStopper - a web-based decision aid for deprescribing in older adults: a protocol 

Dear Dr. Monteiro:

I'm pleased to inform you that your manuscript has been deemed suitable for publication in PLOS ONE. Congratulations! Your manuscript is now with our production department. 

Kind regards, 

on behalf of

Dr. Simon White 

Academic Editor

PLOS ONE